# A Systematic Review on the Cognitive Benefits and Neurophysiological Correlates of Exergaming in Healthy Older Adults

**DOI:** 10.3390/jcm8050734

**Published:** 2019-05-23

**Authors:** Robert Stojan, Claudia Voelcker-Rehage

**Affiliations:** Department of Human Movement Science and Health, Chemnitz University of Technology, Thueringer Weg 11, DE-09126 Chemnitz, Germany; robert.stojan@hsw.tu-chemnitz.de

**Keywords:** exergaming, exercise, aging, healthy older adults, cognitive functions, neurophysiological correlates

## Abstract

Human aging is associated with structural and functional brain deteriorations and a corresponding cognitive decline. Exergaming (i.e., physically active video-gaming) has been supposed to attenuate age-related brain deteriorations and may even improve cognitive functions in healthy older adults. Effects of exergaming, however, vary largely across studies. Moreover, the underlying neurophysiological mechanisms by which exergaming may affect cognitive and brain function are still poorly understood. Therefore, we systematically reviewed the effects of exergame interventions on cognitive outcomes and neurophysiological correlates in healthy older adults (>60 years). After screening 2709 studies (Cochrane Library, PsycINFO, Pubmed, Scopus), we found 15 eligible studies, four of which comprised neurophysiological measures. Most studies reported within group improvements in exergamers and favorable interaction effects compared to passive controls. Fewer studies found superior effects of exergaming over physically active control groups and, if so, solely for executive functions. Regarding individual cognitive domains, results showed no consistence. Positive effects on neurophysiological outcomes were present in all respective studies. In summary, exergaming seems to be equally or slightly more effective than other physical interventions on cognitive functions in healthy older adults. Tailored interventions using well-considered exergames and intervention designs, however, may result in more distinct effects on cognitive functions.

## 1. Introduction

As a result of demographic change, modern societies are facing a progressively growing proportion of older adults over the next decades [1]. Based on this development, there will be a correspondingly higher number of elderly people suffering from age-related cognitive decline and (corresponding) neurodegenerative diseases (NDs) like Parkinson’s disease (PD) [2] or Alzheimer’s disease (AD) [3]. This will inevitably lead to further implications for social, occupational and health care systems [4,5,6].

Despite large inter- and intra-individual differences, mainly fluid cognitive functions (e.g., executive functions, attention, visuospatial skills, processing speed) show essential declines with age, while crystalline abilities (e.g., language or vocabulary, arithmetical skills, or general knowledge) seem to remain relatively stable and show little deteriorations, if at all [7,8,9,10]. The decrease in cognitive performance is associated with a variety of age-related brain changes, including the shrinkage in grey and white matter volume [11,12,13], region specific changes in brain connectivity [14,15], losses in brain vascularization and cerebral blood flow [16,17], higher rates of neuroinflammation [18,19], declining levels of neurotrophins [20,21], and dysregulations in neurotransmitter systems [22,23]. Those brain changes (especially volumetric changes) are most prominent in anterior regions that are particularly important for information processing and strongly associated with fluid cognitive functions [24,25,26]. Evident changes in other brain regions, such as the temporal and parietal lobe, subcortical areas (particularly the hippocampus), or cerebellum, are less distinct, but undoubtedly contribute to cognitive aging as well [9,11,13,27,28].

Based on the structural and metabolic changes, brain function is typically altered in aging individuals. Although study findings seem to be quite heterogeneous, characteristic differences in brain activation have been reported between younger and older adults. For example, while performing cognitive tasks, older adults might display (1) a shift in activation from posterior to anterior brain regions (PASA phenomenon) [29,30], (2) higher or lower (pre-)frontal activation depending on, for instance, the difficulty and type of task [31,32,33], and (3) a reduction in hemispherical lateralization (HAROLD effect) [34]. These age-specific functional patterns are discussed, for example, within the ‘compensation related utilization of neural circuits’ (i.e., CRUNCH) model [32,35,36,37]. Insufficiently functioning or dysfunctional brain regions are hereby supported by usually not involved brain areas in order to maintain cognitive and behavioral performance. Supporting this compensational view of age-related brain adaptations, other cognitive aging models, such as the STAC-r model (Scaffolding Theory of Ageing and Cognition) [7,38], integrate evidence of potential beneficial and detrimental factors, such as lifestyle factors, social/intellectual engagement, or exercise approaches, influencing the efficacy of compensational networks. The variety of those factors might help to explain individual differences and heterogeneous study findings. Furthermore, such models might contribute to develop effective multi-domain lifestyle and intervention strategies to maintain cognitive performance and brain health up to old age.

A variety of non-pharmaceutical intervention concepts have been developed and evaluated to attenuate age-related cognitive decline or to even improve cognitive performance in older adults. Apart from different lifestyle approaches (e.g., nutrition or education), particularly physical exercise [39,40,41,42] and cognitive training [43,44,45,46] have been shown to benefit cognitive and brain health [38,47,48]. Physical exercise appears to induce physiological and metabolic changes that in turn facilitate particular cognitive functions (specifically executive functions) through brain structural and functional adaptations [39,40,41,49,50]. In contrast, cognitive training appears to benefit the trained cognitive abilities almost exclusively with very limited transfer to untrained domains [44,45,51,52]. Thus, both physical and cognitive training, vary regarding their effects on cognitive and brain function [53,54,55]. Both forms of intervention, however, seem to depend on, for example, training intensity, duration, frequency, and type of exercise. Potential moderators of training efficacy, however, still need to be determined more precisely [40,44,45,55,56,57].

Based on the individual beneficial effects of physical and cognitive training, combined interventions have been developed to maximize training efficiency and cognitive benefits [58]. So far, review articles concluded that combined physical and cognitive training interventions show larger effects on cognitive functions than single-domain physical or cognitive training and even subsequently applied physical and cognitive training [59,60,61]. In this vein, exergaming has become an interesting approach. Exergaming, as a novel form of exercise, likewise combines physical and cognitive exercise in an interactive digital, augmented, or virtual game-like environment. Commercial exergame systems, such as the Nintendo Wii, Xbox Kinect, or Dance Dance Revolution have already been demonstrated to be equally demanding as moderate physical exercise [62,63,64,65]. Furthermore, playing video games seems to be beneficial for cognitive functions and therefore might substitute (non-)computerized cognitive training [33,66]. Thus, combining physical exercise and gaming in exergames might be a promising approach to facilitate cognitive and brain functioning in older adults.

A variety of systems have been utilized for exergame interventions, all of which were summarized to have equal or superior benefits on cognitive functions than single-domain and combined physical-cognitive training [67,68]. Exergames have been discussed to amplify the effects of physical exercise by guiding neuroplastic changes via additional cognitive exercise [69,70,71]. However, literature is lacking an updated and comprehensive systematic review on cognitive benefits and the underlying neurophysiological mechanisms of the complementary physical-cognitive training effects of exergaming in healthy older adults. Therefore, we performed an extensive systematic literature search to gather evidence on intervention effects of exergaming on cognitive domains with a particular focus on neurophysiological outcomes. We will confer our findings with respect to the current literature on physical, cognitive, and combined exercise interventions. Moreover, we will discuss potential neurobiological mechanisms underlying exergaming effects and methodological considerations for prospect exergame intervention studies.

## 2. Materials and Methods

### 2.1. Search Strategy

The procedure of this systematic review followed the PRISMA guidelines [72]. Two independent researchers conducted an electronic search between September 2017 and June 2018 in the following databases: Cochrane Library, PsycINFO, Pubmed and Scopus. The last search update was performed on June 5, 2018. Based on a previous exploration of relevant publications, our search was limited to studies published from January 2008 to present in English or German language. Terms of our determined search string were connected by “AND” and “OR” operators. To ensure high sensitivity of our literature search, we performed a title, abstract and key word search applying the following search string individually adjusted for each data base: (old* OR elder* OR aging OR ageing OR “60 year*” OR senior*) AND (exergam* OR Wii* OR Kinect* OR “*active video gam*” OR AVG* OR “computer* gam*” OR “virtual reality” OR computerized) AND (cogniti* OR executive OR “reaction time*” OR memor* OR “processing speed” OR attention* OR “perceptual speed” OR verbal OR “dual task*” OR “dual-task*”).

### 2.2. Selection Criteria

For the purpose of our systematic review, we considered exergaming as physical activity in an interactive and cognitively demanding digital, augmented, or virtual game-like environment. Activities in sitting conditions usually controlled entirely with a computer mouse, joystick, gamepad, or similar handheld devices were not considered exergaming, e.g., [73]. We only included trials on healthy older adults (≥60 years of age) without major physiological, cognitive, or psychological conditions and trials that were published in peer-reviewed journals. Further, we only included studies on mixed healthy and non-healthy (e.g., mild cognitively impaired/demented) samples if results for healthy subsamples were reported individually. Studies were eligible if at least one intervention group performed multiple weeks of training with a minimal cumulative duration of four hours comprising any form of exergaming according to our definition. Primary or secondary outcome measures had to capture at least one of the following domains: (1) cognitive functions, (2) measures of cognitive state (e.g., Mini-Mental State Examination (MMSE), Montreal Cognitive Assessment (MoCA)), or (3) brain functional or structural data. We excluded dissertations, case studies, conference papers, or studies that did not investigate any outcomes of interest. We only included studies that investigated the effects of an exergame intervention without additional interventional components (e.g., additional cognitive or physical training, or dietary), unless a similar active control group was applied, whereby exergaming components were omitted or replaced by traditional training activities. Hence, we excluded multidomain interventions (e.g., exergaming plus additional cognitive training or lifestyle intervention) that did not apply respective control groups (exergaming only or all but exergaming) allowing certain inferences on exergame efficacy, e.g., [58,74]. Studies that used the same samples were considered once within the review [75,76].

### 2.3. Selection Process and Data Extraction

After removing duplicates and scanning titles and abstracts, potential studies were screened for relevance by two independent researchers according to our prior set inclusion and exclusion criteria. Remaining studies were discussed for eligibility. A third independent reviewer was consulted for not unequivocal decisions between main reviewers. Additional studies found in the reference lists of the screened publications were also considered and discussed for eligibility (cf. Figure 1 for the flow chart).

All included studies were scanned for relevant information according to our predetermined main study characteristics (First Author, Sample, Study Design, Intervention, Measures of Exercise Intensity, Outcome Measures, Results and Risk of Bias). Significant findings on outcomes of interest are presented in the main table, comprising within group improvements in the intervention and active/passive control groups as well as interaction effects between the intervention, active, and passive control groups. Non-significant findings were not included in Table 1, but presented in Table 2. The number of applied cognitive tests per study was separately recorded to give a better estimate of potential errors introduced by applying multiple tests. Effect sizes were only included if explicitly reported in the study.

### 2.4. Categorization of Cognitive Outcome Parameters

Given the heterogeneous theoretical models of cognitive domains used to classify cognitive functions in the individual studies and respective assignments of cognitive measures, we introduced a standardized categorization scheme of nine cognitive domains: cognitive tests that primarily targeted cognitive abilities, such as working memory/updating, shifting/mental flexibility, inhibition, planning, reasoning, or problem solving, were summarized as measures of executive functions [50,77]. Memory tasks were subdivided and assigned to either short- or long-term memory [78] depending on whether information was held in mind only for a few seconds (e.g., digit span forward, short-term) or at least a few minutes (e.g., story recall, long-term). If information was manipulated while held in mind for a few seconds, tasks were categorized as working memory and attributed to executive functions (e.g., n-back task or digit span backwards). Based on the suggestions of Colcombe and Kramer [42], tasks were ascribed to processing speed if speed of cognitive information processing was measured while no or very low additional neurophysiological resources were utilized (e.g., simple reaction time tasks or digit symbol substitution test). Correspondingly, cognitive tasks that similarly focused mainly on the speed of cognitive information processing, but simultaneously afforded additional cognitive demand, such as a previous or simultaneous differentiation of multiple stimuli, were categorized as controlled processing (e.g., choice reaction time tasks or divided attention tasks). Cognitive measures that primarily assessed participant’s ability to remember or manipulate visual and spatial information were determined as visuospatial skills (e.g., spatial span test or mental rotation). Tasks that examined subject’s language skills, such as an enumeration of words within a specific category (e.g., COWAT), were defined as verbal fluency. Dual-tasking comprised tasks that demanded the simultaneous processing of two distinct but combined continuous motor-cognitive tasks, such as treadmill walking and n-back task. Finally, clinical cognitive screening tests, such as the MoCA and MMSE, were comprised as measures of cognitive state, but listed individually within the tables. According to our categorization scheme, we attributed some cognitive measures to different cognitive domains than the authors of the respective studies suggested. For instance, Maillot, et al. [79] assigned the digit symbol substitution test as a measure of executive functions, while we referred to it as a measure of processing speed. Likewise, measures of attention were either attributed to processing speed or controlled processing.

### 2.5. Assessment of Methodological Quality

Methodological quality of the included studies was assessed by utilizing the ‘risk of bias tool’ provided by the Cochrane Collaboration [80]. The risk of bias tool comprises evaluations of (1) random allocation sequence, (2) allocation concealment, (3) blinding of participants and personnel, (4) blinding of outcome assessment, (5) incomplete outcome data, (6) selective reporting, and (7) other sources of bias. Estimations on risk of bias for each study are presented in Table 1.

## 3. Results

### 3.1. Search Results

A total of 4536 studies were identified within our initial electronic data base search, including Scopus, Pubmed, Cochrane, and PsycINFO. After removing duplicates (*n* = 1827) and after title and abstract screening (*n* = 2636 removed), 73 studies were scanned for full text, whereof 13 studies met our eligibility criteria (*n* = 61 removed: non-relevant outcomes = 24, non-exergaming intervention = 11, non-normative sample = 25, same participants = 1). Further, two eligible studies were found in the reference lists and added to our bibliography. In total, we included 15 studies in our systematic review that investigated the influence of exergame training on either cognitive functions, cognitive state, or neurophysiological parameters in healthy older adults. For a comprehensive flow chart of our search process, see Figure 1. In the following section we detail studies with regard to particular study characteristics, such as study design and participant characteristics, intervention characteristics and types of exergames, and cognitive and neurophysiological outcome parameters of interest.

### 3.2. Study Design and Participant Characteristics

The main study characteristics are summarized in Table 1. All included 15 studies were published between 2010 and 2018 (most frequently between 2015 and 2016, *n* = 7) and conducted in 10 different countries (Switzerland = 3; Australia = 2; Brazil = 2; USA = 2; France, Germany, Taiwan, Japan, Korea = 1; USA-Ireland = 1, for more detailed information, see Table 1). Twelve studies were carried out as randomized controlled trials (RCT) [76,79,81,82,83,84,85,86,87,88,89,90], including one pilot study [83]. Two studies were performed as unrandomized controlled trials (UCT) [91,92] and one study was uncontrolled [93]. Fourteen studies included control groups, five of which only introduced a passive control group that did not receive any form of intervention (i.e., not more than one training session) [76,79,87,88,90] and seven studies introduced active control groups only [81,82,84,85,86,89,92], whereof one study applied two active control groups with and without additional cognitive demand (single- and dual-tasking) [84]. One study administered both, an active and a passive control group [91] and one study compared a high cognitive workload exergame intervention with a low cognitive workload exergame intervention [83].

Participant recruitment took place in local communities, senior living centers/residences, retirement communities/villages, community centers and senior universities. According to our inclusion criteria, only healthy older adults were involved. Participants’ mean age ranged from 60 to 85 years. The proportion of female participants was higher in 12 studies, ranging from 53–100%. A total of 750 participants were recruited among the 15 studies, whereof 405 were assigned to exergame interventions and 345 to control groups (206 active, 139 passive). Group sizes ranged from 14 to 47 participants, most frequent between 14 and 36 (*n* = 13 studies). Schoene et al. [76] and Anderson-Hanley et al. [81] included sample sizes of 47/43 and 38/41, respectively.

### 3.3. Intervention Characteristics and Types of Exergames

Training interventions lasted 11 weeks on average (range: 6–26 weeks) and training doses varied from 40 min/week for 6 weeks up to 120 min/week for 26 weeks. All studies used training durations of 6–12 weeks, except Eggenberger et al. [84] and Schoene et al. [76] who conducted 26 and 16 weeks of intervention, respectively. On average, training sessions were performed 2.4 times/week (range: 1–5 sessions/week) and lasted 40 min (range: 20–80 min). In 12 studies, training was supervised by trainers or physical therapists [79,81,82,84,85,86,87,88,89,91,92,93], while in three studies participants were asked to train on recommended training doses on their own [76,83,90]. Training adherence was reported by self-reports and/or by the exergame software.

Overall intervention adherence ranged from 40% [83] to 100% [88], most often between 80% and 100% (*n* = 11 studies). Lowest adherence rates within exergamers were found in the uncontrolled trial of Studenski et al. [93] with 70% adherence and in one unsupervised study on two exergame groups of Barcelos et al. [83] with 40% and 43% respectively. If non-exergame control groups were included, adherence rates of the control groups were higher in six studies (passive = 3) [76,87,90], (active = 3) [81,84,89], lower in three studies [85,86,92], equal in three studies (passive = 2) [79,88] (active = 1) [82], and not specifically reported in one study [91]. Thus, adherence rates were quite similar between exergamers and control groups. 

The most frequently utilized type of exergame systems among the included studies were dance and step video games (*n* = 7 studies) [76,84,85,89,90,91,93], where participants stood on a platform or floor mat and had to react to visual cues (e.g., arrows) presented on a screen with corresponding foot movements in equivalent directions. Five studies used commercial home video game consoles (Nintendo Wii = 1, Xbox Kinect = 4) [79,82,86,87,92], whereby participants played a variety of interactive video games. In two studies, participants trained on stationary bicycles with different interactive virtual components, including following a scenic bike path, challenging a virtual competitor, or collecting colored coins corresponding to colored dragons [81,83]. Finally, one study [88] applied a virtual kayak program, where participants had to paddle through a 3D virtual environment sitting in a kayak ergometer and requiring visuospatial processing.

Exercise programs for non-exergame active control groups comprised stationary cycling (*n* = 2) [81,86], treadmill walking (*n* = 3) [84,86,91], conventional physiotherapy (*n* = 1) [82], and different forms of coordination, flexibility, stretching and strength exercise (*n* = 3) [85,89,92]. Training frequency and duration were matched for exergamers and active control groups in all respective studies. Physical exercise intensity was controlled for (intensity was adaptively adjusted based on objective parameters) in four studies via measures of heart rate [81,83,86,91] and monitored (intensity was only assessed, but not adaptively adjusted), but not controlled, in three further studies likewise by measures of heart rate [79] or RPE (rating of perceived exertion) assessment [79,84,85]. However, one study [84] monitored exercise intensity only for control groups but not for exergamers and another [86] controlled intensity in their active control group, but only monitored intensity in the exergame group. Furthermore, one study [87] assessed heart rate throughout training sessions before and after each exergame to prevent overexertion. Seven studies did not explicitly report any measure of exercise intensity [76,82,88,89,90,92,93]. Therefore, an overall comparison of exercise intensities between exergamers and active control groups is not possible. Cognitive training intensity was assessed only in one study (i.e., perceived cognitive demand of the training) via RPE rating [85].

### 3.4. Cognitive Outcome Measures

Based on our categorization of nine cognitive domains, the number of tested domains within studies ranged from one to five (for neurophysiological parameters, cf. 3.7). Three studies examined only one outcome of interest [82,88,93], four studies each examined two [83,87,89,91] or four domains [79,84,86,90], and two studies each investigated three [85,92] or five relevant cognitive domains [76,81]. Cognitive functions in the respective domains were assessed by at least one test in all studies, but were most often addressed by multiple measures for the respective function. Executive functions (inhibition, mental flexibility and working memory) were most frequently investigated (*n* = 12 studies) and measured either by use of multiple tests (*n* = 10) [76,79,81,83,84,85,86,87,89,92] or single tests for one specific function (*n* = 2) [90,91]. All studies that assessed executive functions further recorded one to four additional cognitive domains: processing speed (*n* = 9) [76,79,81,84,85,86,87,90,92], controlled processing (*n* = 4) [76,79,89,90], visuospatial skill (*n* = 3) [76,79,81], short-term memory (*n* = 3) [81,84,86] and long-term memory (*n* = 1) [84], dual tasking (*n* = 2) [76,90], verbal fluency (*n* = 2) [81,92], and measures of cognitive state, (*n* = 4, MMSE = 2, MoCA = 2) [83,85,86,91]. For more detailed information on cognitive outcome measures, see Table 1.

### 3.5. Intervention Effects on Cognitive Outcomes

All studies, except one [93], reported positive effects of exergame training on cognitive functions or state (for results on neurophysiological parameters cf. 3.7). Positive effects were referred to by either reporting significant group by time interactions (*n* = 8 studies) in favor of exergamers over passive (*n* = 4) [76,79,88,90] or active control groups (*n* = 4) [81,83,85,92], and/or by reporting improvements within the group of exergamers in at least one cognitive domain (*n* = 8) [81,82,84,86,87,88,89,91]. Most studies that reported a significant group by time interaction, however, found no performance improvements on respective functions within the exergaming group, but at least no change or less decline as compared to the control groups (*n* = 6).

Significant within group improvements and interaction effects between exergaming and control groups varied extensively between and also within studies. Most studies (*n* = 11) reported significant effects of exergaming on one or just a few cognitive outcomes that have been addressed, but not on all outcome measures (for detailed information on the effects on particular functions, see Table 1 and Table 2). Further, there seems to be no coherence across studies for which cognitive domain positive effects were reported. For example, while Anderson-Hanley et al. [81] and Maillot et al. [79] found significant interaction effects (one trend at *p* = 0.07) on all measures of executive functions in favor of exergamers (*n* = 3 and *n* = 5 measures, respectively), Schoene et al. [90], Schoene et al. [76], and Ordnung et al. [87] found no intervention effect on any executive function at all (*n* = 1, *n* = 3, and *n* = 2 measures, respectively). Furthermore, some studies used a battery of cognitive tests comprising different cognitive domains and either found significant effects on the majority of the applied cognitive tests, e.g., [79,84] or only significant results for single cognitive tests [76,81,85,92]. Interestingly, those three studies that specifically controlled exercise parameters between exergamers and active controls (type of physical activity (e.g., cycling vs. cybercycling), training duration, frequency, and intensity) all reported group by time effects in favor of exergamers on at least but not all on executive function [81,83,92], but not consistently on other cognitive domains. Interestingly, we observed no evident differences between studies that compared exergaming to cardiovascular exercise and studies that compared it to forms of balance, stretching and flexibility training.

Intervention effects also did not systematically differ with respect to the type of exergame interventions. Findings on particular cognitive domains varied vastly even within similar types of exergames (for detailed information, see Table 2). With respect to dance and step video game interventions (*n* = 7), three studies observed improvements within exergamers on executive functions (and one additionally for long-term memory and processing speed [84]), but no interaction effects compared to passive or active controls [84,89,91]. One study only reported a group by time interaction for one measure of executive functions in favor of exergamers over active controls, but no within group improvements in exergamers on any measure of executive functions [85]. Two further studies found no significant results on executive functions at all [76,90]. Likewise, studies on commercial home video game consoles (*n* = 5) were quite heterogeneous regarding their effects on cognitive outcomes. Maillot et al. [79] found interaction effects for all of their measures of executive functions (*n* = 5) and processing speed (*n* = 4) and for 1/4 measures of visuospatial skill, but no within group improvements. Kayama et al. [92], however, reported interaction effects only on 1/2 measures of executive functions. Additionally, Guimaraes et al. [86] found within group improvements for only 1/2 measures of executive functions (and short-term memory), and Ordnung et al. [87] found no effects on executive function at all (0/2), but observed within group improvements for exergamers on two measures of processing speed that were non-significant in the other studies. However, contrary study results within types of exergames might have been most probably superimposed by other moderators, such as intervention, sample, or training characteristics.

### 3.6. Potential Moderators of Intervention Effects

Regarding sample characteristics (e.g., age, gender distribution, sample size) and training parameters (e.g., training frequency, duration, intensity), we found no distinct evidence for potential moderating effects on intervention efficacy across studies. However, we have some evidence that single studies might have been influenced by age or training parameters. For instance, in contrast to Anderson-Hanley et al. [81], Barcelos et al. [83] observed interaction effects, but no within group improvements on executive functions in exergamers, although both studies used the same exergame system (cybercycling). This contradiction might be at least partly attributed to a higher mean age of the sample in the Barcelos et al. [83] study (mean age = 85.05 years) or different training parameters (lower dose). Furthermore, we can only speculate about dose-response effects across studies as most studies used intervention periods of up to 12 weeks with varying training durations, while only two studies conducted longer interventions. Interestingly, all three intervention groups in the study of Eggenberger et al. [84] that received 26 weeks of training (2 × 60 min per week) showed cognitive performance improvements in almost all tests (in eight out of nine cognitive tests, comprising measures of executive functions, short-term and long-term memory, and processing speed), which might be attributable to the higher training dose, respectively.

In summary, intervention effects on a variety of cognitive functions were observed within and between groups, but were not systematically found for certain cognitive domains, types of exergames, or particular training characteristics. No study reported cognitive decrements within exergamers or inferior cognitive performance of exergamers compared to passive or active control groups. Potential contributors to exergame intervention success remain widely unclear as study designs as well as sample and training characteristics largely varied.

### 3.7. Neurophysiological Correlates of Exergame Training

Four studies examined neurophysiological outcomes, whereby all studies focused on different parameters. Three studies addressed brain functional parameters and were conducted on dance and step exergames [85,89,91] and one study investigated changes in neurotrophic factor levels of BDNF (brain derived neurotrophic factor) in response to cybercycling [81].

Eggenberger et al. [85] investigated the hemodynamic activity in the prefrontal cortex (PFC) during preferred and fast treadmill walking via functional near-infrared spectroscopy (fNIRS). After 8 weeks of intervention they observed reductions in hemodynamic activity during walking with domain and time specific group by time interaction effects and respective advantages of exergamers (dance video game) over active controls (balance and stretching) at post-test. They associated decrements in brain activity after training with higher efficiency of specialized neural brain networks and lower utilization of compensational networks. Schättin et al. [89] focused on event-related oscillations (ERO: 1–32 Hz) over the PFC during a divided auditory-visual attention task, measured using electroencephalography (EEG). Accordingly, they showed a performance related reduction in theta-band activity at post-test after eight weeks of intervention (3 × 30 min per week) that was only present within exergamers (dance video game), but not within an active control group (balance training). No interaction effects or further differences between groups and for other frequency bands were observed. According to the authors, their results reflect an adverse aging effect that is often but not consistently indicated by higher theta-band power in older compared to younger individuals and might therefore indicate improvements in efficiency of specialized neural networks. Chuang et al. [91] applied EEG during a flanker task and analyzed event related potentials (ERP: N2/P3, latency and amplitude) over the fronto-parietal cortex. They found shorter RTs within exergamers (dance video games) and active controls (brisk walking) and longer RTs in the passive control group after 12 weeks of intervention (3 × 30 min per week), but no significant interaction effect between groups on RTs. However, they found a significant interaction between exergamers/active controls and passive controls for N2 and P3 latency, while both active groups exhibited shorter N2/P3 latencies during the flanker task. As higher N2 and P3 latencies are associated with aging, lower latencies may reflect higher processing efficiency of specialized networks [91]. Notably, they did not observe any differences between exergamers and active controls. Finally, Anderson-Hanley et al. [81] investigated BDNF levels pre and post intervention (not directly after exercising) that were quantified using ELISA (enzyme-linked immunosorbent assay). They reported significant group by time differences on BDNF levels, while cybercyclers exerted higher levels of BDNF compared to a traditional cycling group after 16 weeks of intervention (5 × up to 45 min). Their findings suggest a more pronounced capability of neuroplastic adaptions in response to exergaming as compared to single domain physical training.

In conclusion, all studies that integrated neurophysiological assessments found evidence on neuroplastic changes following exergame training and primarily referred them to improvements in specialized neural networks and/or a lower dependency on compensational support. Interaction effects on neurophysiological parameters in favor of exergamers over passive [91] and active [81,85,89] control groups were present in all studies (cf. Table 1 for an overview of the intervention effects on particular neurophysiological outcome parameters).

### 3.8. Methodological Quality

Regarding risk of bias assessment [80], we found that three studies had a high risk of introducing sources of bias [81,83,91], four a moderate risk [84,85,89,93], six a low risk [76,82,86,87,88,90], and for two studies risk of bias could not be assessed because of lack of information, termed as unclear [79,92]. The most frequent risk of bias was introduced by insufficient randomization and blinding procedures.

## 4. Discussion

In this systematic review, we present an updated and comprehensive overview of the effects of exergaming on cognitive and brain function in healthy older adults. We found 15 studies that addressed the effects of exergame training on cognitive functions or cognitive state, four of which included brain functional parameters. Overall, exergaming has been shown to yield very inconsistent benefits only on specific cognitive functions (mainly executive functions) and appeared to be approximately equally beneficial as compared to other forms of physical exercise. We observed a vast variability of intervention and training designs and no coherence of outcome effects on cognitive domains. Neurophysiological changes with regard to exergaming (within exergamers or by group × time effects) were present in all corresponding studies (either on hemodynamics, electrophysiology, or neurotrophic factors) indicating brain plastic adaptations in response to exergaming. In summary, we conclude that exergaming might have the potential to maintain or facilitate cognitive and brain health in healthy older adults and might be recommended in addition or in substitution to traditional forms of exercise. However, beneficial factors contributing to intervention efficacy as well as the neurophysiological mechanisms underlying exergaming need to be investigated more systematically.

### 4.1. Types of Exergames

Single-domain training interventions, such as cardiovascular or cognitive training, revealed that cognitive benefits seem to depend on the type of physical activity (aerobic, resistance, and coordination training) [39,40] as well as on the primarily afforded cognitive domains during cognitive training [44,45]. Accordingly, we expected certain types of exergames to yield distinct effects on cognition based on their individual physical and cognitive demand. Exergames that afford specific cognitive functions might improve these, but might be less beneficial for less demanded functions [69,70]. Moreover, cognitively higher demanding exergames likewise might yield larger effects than cognitively less effortful exergames. Interestingly, we found no such consistencies between exergame systems regarding intervention effects on cognitive domains. In contrast, we even observed varying effects within all different categories of exergames. Here, we are in line with earlier reviews and meta-analyses on exergaming [61,67,68,94,95,96].

Further, no study except Barcelos et al. [83] systematically varied the cognitive load during exergaming while keeping exercise intensity, type, frequency and intervention duration constant between groups or vice versa. The latter reported a significant group by time interaction effect on Stroop performance, but not for other executive functions, in favor of exergamers that trained with a higher demanding and cognitively more specific exergame (cycling while collecting coins corresponding to colored dragons). Controls trained with a lower demanding and non-specific exergame (cycling plus virtual bike tour). Those results indicate that exergames may be used as a potentially tailored intervention. The cognitive demand during exergaming might lead to greater improvements in those cognitive domains and corresponding brain networks that are more active during exergaming. One reason why we found such results only in one study might be that potential related effects of similar exergame systems were superimposed by different intervention designs and training characteristics that varied considerably across studies. Consequently, further research is required to validate the potential guidance of cognitive improvements of exergames.

As mentioned above, types of exergames can be roughly divided into three distinct categories: (1) dance and step video games, (2) commercial home video game consoles and (3) interactive virtual ergometers (cybercycles and virtual kayak ergometer). Different exergame systems most likely varied regarding their physical and cognitive demands [63,64,65]. However, as most studies did not systematically report and control physical and cognitive demands, we can only speculate about the distinct demands of the different exergame systems. On the one hand, commercial consoles, like the Nintendo Wii or Xbox Kinect, provide an extensive set of different exergames and, respectively, varying forms of physical activity and cognitive demands [62,63], whereas virtual ergometers, dance video game platforms, or step mats may provide restricted physical-cognitive training parameters based on the lower game variety and similar forms of physical activity and cognitive demand [97,98]. On the other hand, exercise intensity may be easier to control with virtual ergometers or dance and step systems by manipulating stimulus presentation time or presentation frequencies (e.g., based on measures of heart rate), which ensures a relatively stable and individually adjustable physical and cognitive intensity level, e.g., [81,83,90]. In contrast, commercial video game consoles exert varying and interval like exercise demands depending on the type of game, individual breaks, and game success. Conclusively, exergame systems might considerably vary in their physical and cognitive demand [62,63,99,100]. This, however, is not reflected in the results on cognitive outcomes.

### 4.2. Intervention Effects on Cognitive Outcomes

A total of 14 out of 15 studies reported beneficial effects of exergaming on cognitive domains. Positive findings referred to either within group improvements within exergamers or group by time effects in favor of exergamers over passive/active controls. Positive effects, however, differed greatly across studies and showed no obvious coherence. For example, while five studies reported within group improvements for exergamers on executive functions, seven other studies failed to replicate that same effect. Similar inconsistencies were observed for group by time effects and other cognitive domains leading to the assumption that exergaming might have no or very small and inconsistent benefits on distinct cognitive functions (cf. Table 2). Besides these heterogeneous results, it should be noted, that no study reported performance decrements within exergamers or poorer performance of exergamers compared to active/passive control groups after intervention. Therefore, our findings are in line with previous reviews [61,94,95,96] and meta-analyses [67,68] on exergaming, even though we included at least seven additional studies [67]. The versatile research body on exergaming might result from several reasons, including: (1) the potentially different physical and cognitive demand of particular exergame systems, (2) varying intervention and training characteristics or (3) insufficient statistical analysis not controlling for multiple testing to avoid false positives results within a multitude of applied tests. Furthermore, (4) some studies were not primarily interested in cognitive but rather in physiological parameters and may have applied cognitive tests that are less sensitive to exercise induced changes (e.g., measures of cognitive state). Further studies systematically investigating and controlling the effects of different training and exergame modalities are needed in order to create a clearer picture on how exergaming might be most beneficial for cognitive and brain health in older adults.

When interpreting results on exergaming, it needs to be considered that studies further differed with regard to the applied control groups. Studies that used an active control group participating in “traditional” exercise interventions (cardiovascular activity, balance and stretching training) as compared to exergaming revealed either equal or slightly superior effects in favor of exergaming. Superior effects of exergaming, however, have been only reported for executive functions, but not for other cognitive domains (cf. Table 2), regardless of the type of exergame. This is an important finding as most studies on cardiovascular exercise interventions in older adults also predominantly observe improvements in executive functions that seem to be most sensitive to exercise induced changes [40,42,47]. In this vein, we found that those studies precisely controlling training parameters (type of physical activity, intensity, frequency, duration) for both exergamers and active controls observed significant group by time effects on at least one measure of executive functions in favor of exergamers [81,83,92]. Therefore, exergaming might be able to amplify the effects on executive functions by adding further cognitive demand to physical activity that might help to integrate physically evoked neurophysiological changes in respective brain regions [101]. However, it needs to be considered that executive functions are investigated most frequently within training studies and might therefore be more likely to yield significant effects in general. The similar effects of exergames and “traditional” physical exercise (mostly cardiovascular exercise) on other cognitive domains, however, might indicate that exergames have no further advantages over traditional exercise.

Interestingly, we found no distinct differences between studies that compared exergaming to cardiovascular exercise [81,83,84,86,91] and those that compared it to stretching, balance and flexibility exercise [82,85,89,92]. This finding is quite unexpected as metabolic exercise (aerobic and resistance exercise) usually results in a more pronounced cognitive benefit in older adults than physiotherapeutic interventions or balance and stretching exercise [40,102]; balance and stretching exercise has often been used as active control group in comparison to cardiovascular exercise. However, potential differences between training forms might have been superimposed by varying intervention and training parameters, such as varying physical intensity levels for the different exergame systems as discussed earlier. For example, studies that compared exergaming to cardiovascular exercise might have used physically higher demanding exergame systems and vice versa, which might have diminished the typically higher efficacy of cardiovascular exercise. However, as physical intensity was inconsequently reported, we can only speculate about potential moderating effects.

As expected, comparisons between exergamers and passive controls were more likely to yield significant effects and for a broader range of cognitive domains. As older adults typically show progressive cognitive declines with increasing age [31,103], intervention effects in favor of active participants might be intensified by the opposing directions of cognitive developments. Therefore, even if interventions may have led to only a preservation of cognitive performance and no cognitive improvements, group by time effects may have shown significance as a result of the cognitive decline in passive samples. Correspondingly, passive control groups only provide limited information on intervention efficacy; especially regarding beneficial training parameters. Even though passive control groups eliminate potential test-retest bias, further studies should apply matched active control groups more frequently in order to assess exergame efficacy and beneficial training parameters more specifically.

### 4.3. Potential Moderators of Exergame Intervention Efficacy

As known from the research body on single-domain physical and cognitive training as well as combined training interventions, several factors seem to moderate intervention effectiveness on cognitive outcomes. For example, sample characteristics (age, gender, cognitive impairment, baseline performance) and training parameters (intensity, frequency, duration) have been shown to potentially modify cognitive benefits following exercise interventions [45,55,56,59,60,66]. Bamidis et al. [58] observed that exergaming in combination with other forms of exercise (i.e., physical and cognitive) facilitates cognitive functions particularly in healthy individuals and seems to be less effective in already cognitively impaired persons. Furthermore, they showed that a lower baseline performance is more likely to result in higher training gains than an already higher baseline performance. Although we only included studies on healthy older adults, samples still differed regarding mean age, gender distribution, or cultural group. As already suggested, certain sample characteristics might influence cognitive training gains [58,104] and particularly affect gains in executive functions [50]. However, while we found no consistent influence of particular factors, sample characteristics might have still interacted with other potential moderators (e.g., training characteristics), influencing intervention effects. For example, sedentary or old older adults (above 80 years of age) might already show cognitive benefits from few training sessions with relatively low physical-cognitive demand [105], because of their lower baseline performance. In contrast, physically and cognitively fit or younger older adults might need more training sessions and higher physical-cognitive demands in order to increase cognitive performance as a result of potential ceiling effects and less impaired brain functionality.

Correspondingly, cognitive benefits through exercise seem to be modulated by different training characteristics, such as exercise intensity, duration, or frequency, although findings likewise vary [55,56,67]. Within our systematic review, study findings varied strongly with regard to intervention length (6–26 weeks), training frequency (1–5 sessions per week), and training duration (20–80 min per session). However, training characteristic showed no obvious and systematic influence on intervention efficacy (e.g., dose-response effects). This has been shown in the meta-analysis by Stanmore et al. [67] as well. They found only a longer intervention length (>12 weeks) to be more effective, while optimal training duration, frequency, and intensity still need to be evaluated more precisely. One reason for such different study designs might be that up to now no training recommendation for exergaming exists to improve cognitive functions in older adults. Likewise, recommendations for single-domain and combined interventions are vague [45,55,59,61].

Even though we did not identify particularly effective training parameters, based on the current results, we suggest that exergaming should be performed two to three times per week for 45–60 min for at least 12 weeks to improve cognition in healthy older adults, partly based on the suggestions of Lauenroth et al. [59] and Tait et al. [61] for combined physical-cognitive training. However, it is important to adjust training parameters adaptively to individual training gains, but also to specific samples (e.g., patients with dementia or PD) to avoid physical as well as cognitive overexertion that could probably lead to detrimental effects as well [45,106]. For instance, if either physical intensity or cognitive demand during exergaming are too challenging (in particular for clinical patients), training motivation and adherence might suffer and reduce intervention efficacy, respectively. To further amplify exercise effects on cognitive functions, it could also be considered to add non-exercise components (e.g., dietary) and brain stimulation techniques (e.g., transcranial direct current stimulation) to exergame interventions [107,108].

### 4.4. Neurophysiological Correlates of Exergaming

Four studies presented evidence on neurophysiological adaptations following exergame training for different brain functional domains, either within the group of exergamers or compared to passive/active control groups. Respective findings comprised alterations in electrophysiology [89,91], hemodynamic activity [85] and levels of BDNF [81]. Brain functional changes opposed typical patterns observed in aging individuals. Therefore, neurophysiological changes suggest a positive influence of exergaming towards a more youth-like brain function that might underlie corresponding cognitive benefits.

Schättin et al. [89] reported enhanced behavioral performance in executive functions and controlled processing within exergamers and an equivalent reduction in theta-band power (during controlled processing) that is usually increased in older adults and therefore reflects a more youth like pattern and potentially higher cognitive performance in response to exergaming [109]. However, other frequency bands were not affected and other parameters, such as the theta-alpha ratio, were not reported [110]. Similarly, Chuang et al. [91] found lower ERP latencies (N2/P3) for exergamers and brisk treadmill walkers as compared to waitlist control participants and correspondingly significantly lower reaction times during executive task performance. Lower ERP latencies reflect a faster neuronal processing speed that typically decreases during aging and is associated with reductions in axonal fiber myelination. In this vein, cognitive processing speed is discussed to mediate the effects of exercise on further cognitive functions and particularly on fluid cognitive functions [111]. Processing speed is largely related to white matter integrity [15,112] and axonal fiber myelination [113]. Physical activity has been linked to enhancements in white matter integrity in older adults [114] and myelin sheet regeneration in rodents [115] and, respectively, to improvements in cognitive processing speed [116,117,118]. As opposed to crystalline functions, fluid cognitive functions rely especially on fast information processing and hence axonal conduction velocity, which is why they might particularly benefit from exercise interventions. The findings of lower ERP latencies in Chuang et al. [91] in correspondence with behavioral performance improvements support this view on the moderating effect of processing speed.

### 4.5. Potential Microbiological Mechanisms Underlying Exergaming

The underlying microbiological mechanisms of exergame induced brain functional changes might be related to distinct modulations of different parameters, such as elevated neurotrophic factor levels, most importantly BDNF, IGF-I (insulin-like growth factor) and VEGF (vascular-endothelial factor). BDNF, IGF-1 and VEGF have been shown to be raised by regular physical and cognitive activity and to contribute to brain structural changes by promoting synaptogenesis, neurogenesis and angiogenesis [49,101,119,120,121,122,123,124]. Anderson-Hanley et al. [81] examined changes in BDNF release in response to cybercycling and ergometer cycling (active controls). They found a significantly higher BDNF level in exergamers compared to ergometer cyclers and correspondingly higher behavioral performance in exergames for executive functions, while the performance of the control group decreased in one measure of executive functions. Training frequency, intensity and type of physical exercise were similar for both groups, which is why differences in BDNF levels might be inferred to the additional cognitive demand within cybercyclers. These results coincide with findings that neurotrophic factors are released via a variety of different signaling pathways that might differ for physical and cognitive activity and seem to add up for combined physical-cognitive interventions, such as in exergaming (e.g., cycling + additional cognitive load) [125]. More precisely, different promoter proteins of BDNF (proBDNF), for example, are constructed on distinct gene segments and are activated via different signaling cascades that seem to differ for physical and cognitive activity [126,127]. For combined training, this results in an overall higher level of proBDNF that correspondingly converses to higher levels of the mature BDNF form (mBDNF). In turn, higher mBDNF levels enable greater structural changes on synapses and other cell structures that ultimately lead to more pronounced brain functional and cognitive improvements [128,129]. Next to that accumulation of transcriptional activators, neurotrophic factors further seem to interact with each other. For example, the appearance of BDNF increases productions of IGF-1 and vice versa, while IGF-1 additionally increases the binding affinity of BDNF on corresponding receptors. Respectively, combining different exercise forms (i.e., physical and cognitive exercise) that elevate distinct factors might facilitate brain functional changes through beneficial interactions of present neurotrophic factors, hormones and other microbiological parameters [119,130,131].

Next to elevated neurotrophic factor levels that induce brain structural changes, physical exercise has been shown to promote brain health and function via improvements in brain vascularization and cerebral blood flow [132,133,134,135], reduced neuroinflammation [136,137] and regulated oxidative stress [138], and enhanced neurotransmitter release [139,140,141]. As exergaming likewise includes physical exercise, respective mechanisms might be present as well. However, further research is needed to validate and understand the underlying macro- and micro-biological mechanisms of exergaming.

### 4.6. Methodological Considerations

Most of the utilized exergame systems have been shown to be easy to use, feasible for older adults, and affordable for use at home. Affordable and motivating exergame systems are of particular importance to consequently transfer results from exergaming studies on cognitive and brain health into communities as everyday life training methods. In general, exergaming is particularly characterized by its versatile combination possibilities of different forms of physical activity (aerobic, strength, coordination, or multicomponent exercise) and digital or virtual gaming components. However, this versatility not only enables varied applications, it also leads to low comparability between individual studies. Effects on cognitive and brain function are probably based on the variability of potential moderating factors. To increase comparability between systems and of physical and cognitive training components, exergames should be specifically evaluated regarding their physical effort (e.g., intensity and type) and cognitive demand (amplitude and required functions). While at least some studies monitored physical intensity, cognitive demand was assessed only in one out of 15 studies [85]. Those parameters, however, should be at least monitored by subjective measures of perceived exertion if no objective measures are available to control for moderating effects of different physical and cognitive loads. Further, prospect studies could evaluate the cognitive demand of the exergames by correlating exergame performance with performances on traditional measures of corresponding cognitive functions. This would allow inferences on potential domain specific performance improvements following exergame training that affords particular cognitive functions. Furthermore, participants’ motivation for the use of different exergame systems as well as expectations about the potential intervention benefits should be controlled for in prospect studies, as both parameters are discussed to mediate intervention effects [142,143,144].

Most importantly, to address potential benefits of exergames over traditional exercise forms as well as basic principles and mechanisms underlying exergame interventions, appropriate control groups, exercising physically and cognitively only should be applied more often. Ideally, single-domain control groups are trained similar to exergamers with no or minimized physical or cognitive demand, such as in the studies of Anderson-Hanley et al. [81], Barcelos et al. [83], and Kayama et al. [92]. Further, to identify beneficial training parameters (i.e., intensity, duration, frequency), prospect studies should systematically vary respective parameters. To sum up, the elaboration of moderating factors within the versatility of exergames that contribute to intervention efficacy is crucial and requires methodological designs to be implemented (more) sophistically.

## 5. Conclusions

The intention of this systematic review was to present a comprehensive overview of the efficiency and underlying neurophysiological processes of exergame training in older adults. We found an overall small and strongly varying positive influence of exergaming on cognitive and brain function in healthy older adults. Benefits on individual cognitive domains showed no consistency. Besides these heterogeneous findings, studies that compared exergaming to traditional types of exercise, such as cardiovascular exercise, found similar or slightly superior effects of exergaming on executive functions but not on other cognitive domains. This might be an indicator that exergaming is a promising approach to preserve and facilitate cognitive and brain health in healthy older adults. Based on the variety of exergame systems and corresponding games, exergaming might be a useful alternative to traditional exercise to further motivate (older) people to regularly engage in physical activity. However, further research is urgently needed to determine potential influencing factors that contribute to intervention efficacy in order to understand the underlying mechanisms and to apply tailored intervention for particular target groups and individual needs.

## Figures and Tables

**Figure 1 jcm-08-00734-f001:**
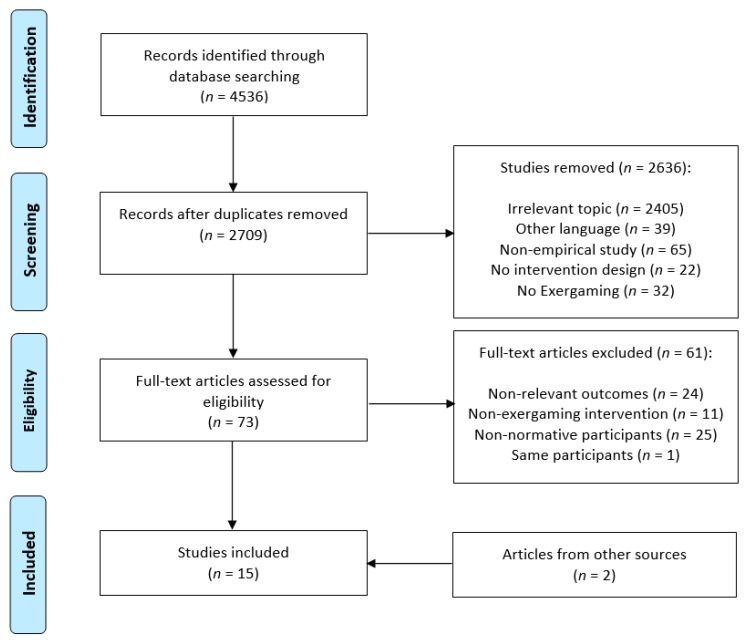
PRISMA flow chart of the search process.

**Table 1 jcm-08-00734-t001:** Study characteristics.

First Author	Sample	Study Design	Intervention	Measures of Exercise Intensity	Outcome Measures	Results	Risk of Bias
Anderson-Hanley et al. (2012) [81]	Pre EG: *n* = 38, Age = 75.7 (9.9), F = 87% CG: *n* = 41, Age = 81.6 (6.2), F = 71% Post (12 weeks) EG: *n* = 30 CG: *n* = 33 R = 58–99 years	USA Two Groups RCT Supervised	12 weeks, 5 × up to 45 min per week recommended, minimum of 25 rides for “completers” EG: stationary cycling with 3D virtual bike tour CG: stationary cycling	Controlled with HRR (60%)	EF (*n* = 3) PS (*n* =1) VF (*n* = 2) STM (*n* = 5) VSS (*n* = 2) BDNF (ELISA)	EG ↑in EF (2/3, *d* = 0.5) CG ↓ in EF (1/3) EG > CG in *EF (*n_p_^2^* = 0.21) EG > CG in BDNF Adherence: EG (79%), CG (80%)	High
Bacha et al. (2018) [82]	Pre EG: *n* = 25 CG: *n* = 25 Post (7 weeks) EG: *n* = 23, Age = 71.0, F = 65% CG: *n* = 23, Age = 66.5, F = 83% Retention (4 weeks after post-test)	Brazil Two Groups RCT Supervised	7 weeks, 2 × 60 min per week EG: Xbox Kinect Adventure Games CG: Conventional Physiotherapy	n.a.	MoCA	EG ↑ in MoCA CG ↑ in MoCA Adherence: EG (92%), CG (92%)	Low
Barcelos et al. (2015) [83]	Pre EGI: *n* = 23 EGII: *n* = 25 Post (12 weeks) EGI: *n* = 10, Age = 84.8 (11.3), F = 25% EGII: *n* = 10, Age = 85.3 (6.9), F = 80%	USA and Ireland Two Groups RCT Pilot Unsupervised	12 weeks, 2 × 20 min per week (minimum threshold, recommended increase 3–5 × up to 45 min per week) EGI: stationary Cybercycling plus video game EGII: stationary Cybercycling plus virtual tour	Controlled with HR	EF (*n* = 3) MoCA	Results on healthy subsamples EGI > EGII (1/3, *n_p_^2^* = 0.69) (Full) Adherence: EG (40%), EGII (43%)	High
Chuang et al. (2015). [91]	Pre *n* = 32, F = 100% Post (12 weeks) EG: *n* = 7, Age = 69.4 (3.8) CGI: *n* = 11, Age = 67.0 (1.7) CGII: *n* = 8, Age = 68.3 (4)	Taiwan Three Groups UCT Supervised	12 weeks, 3 × 30 min per week EG: Dance Dance Revolution CGI: brisk walking CGII: passive	Controlled with HRmax (40–60%), RPE	EF (*n* = 1) MMSE EEG: ERPs during Flanker task	EG/CGI ↑ in EF (*d* = 1.18 (EG) and 0.37 (CGI)) CGII ↓ in EF (*d* = 0.72) N2 latency: EG and CGI < CGII (*d* = 2.02 (EG), d = 2.51 (CGI)) P3 latency: EG and CGI < CG II (*d* = 1.03 (EG), *d* = 1.21 (CGI)) Adherence: overall (81%)	High
Eggenberger et al. (2015) [84]	Pre EG: *n* = 30 CGI: *n* = 29 CGII: *n* = 30 Between (3 Month) EG: *n* = 25 CGI: *n* = 24 CGII: *n* = 26 Post (6 Month) EG: *n* = 24, Age = 77.3 (6.3), F = 58% CGI: *n* = 22, Age = 78.5 (5.1), F = 72% CGII: *n* = 25, Age = 80.8 (4.7), F = 64% Retention (12 month after post-test)	Switzerland Three Groups RCT Supervised (except 4 weeks)	26 weeks, 2 × 60 min per week in group settings, 20 min group specific training (EG, CGI, CGII) + 40 min balance and strength training EG: Virtual Reality Dance Video Game (Platform) CGI: treadmill + stimulus verbal memory training CGII: treadmill + strength and balance exercises for each group	RPE (monitored only for CGs)	EF (*n* = 2) PS (*n* = 4) STM (*n* = 1) LTM (*n* = 2)	EG/CGI/CGII ↑ in *EF EG/CGI/CGII ↑ in *LTM EG/CGI/CGII ↑in *PS Adherence: 3 m 6 m 18 m EG (83%, 80%, 50%), CGI (83%, 75%, 58%), CGII (87%, 83%, 50%)	Moderate
Eggenberger et al. (2016) [85]	Pre EG: *n* = 22 CG: *n* = 20 Post (8 weeks) EG: *n* = 19, Age = 72.8 (5.9), F = 63% CG: *n* = 14, Age = 77.8 (7.4), F = 64%	Switzerland Two Groups RCT Supervised	8 weeks, 3 × 30 min per week EG: Virtual Reality Dance Video Game (Platform) CG: balance and stretching	RPE (monitored)	EF (*n* = 3) PS (*n* = 1) MoCA fNIRS: HbO_2_ level during preferred and fast treadmill walking (PFC)	EG > CG in EF (1/3) EG/CG ↓ PFC activity (first 7 s of preferred walking, r = 0.32/0.36, trend fast walking, *r* = 0.25) EG < CG PFC activity (last 7 s of fast walking *r* = 0.32) EG/CG: lPFC > rPFC activity after training, *r* = 0.26–0.32 EG > CG: lPFC activity (*r* = 0.23/0.27) Adherence: EG (86%), CG (64%)	Moderate
Guimaraes et al., (2018) [86]	Pre *n* = 36, F = 61% Post (12 weeks) EG: *n* = 13, Age = 60.0 (4.0), F = 77% CG: *n* = 14, Age = 60.7 (3.6), F = 43%	Brazil Two Groups RCT Supervised	12 weeks, 3 × 60 min per week (+ 3 previous sessions) EG: Microsoft Kinect Sport Games CG: Treadmill and cycle ergometers Both: 5–10 min warm up and 5–10 min stretching	HR (EG monitored; CG controlled: 40–59% HRR)	EF (*n* = 2) PS (*n* = 2) STM (*n* = 2) MMSE	EG ↑ in EF (1/2) EG ↑ in STM (1/2) CG ↑ in EF (1/2) CG ↑ in PS (1/2) CG ↑ in *STM CG ↑ in MMSE Adherence: EG: (91%), CG (87%)	low
Kayama et al. (2014) [92]	Pre Age ≥ 65EG: *n* = 30 CG: *n* = 18 Post (12 weeks) EG: *n* = 26 CG: *n* = 15	Japan Two Groups UCT Supervised	12 weeks, 1 × 80 min per week EG: 75 min traditional training (cf. CG) + 5 min Dual Task Tai Chi (Kinect) CG: 80 min aerobic, strength, balance, and flexibility training	n.a.	EF (*n* = 2) PS (*n* = 1) VF (*n* = 1)	EG > CG in EF (1/2) Adherence: EG (87%), CG (83%)	Unclear
Maillot et al. (2012) [79]	Pre F = 84% EG: *n* = 16 CG: *n* = 16 Post (12 weeks) EG: *n* = 15, Age = 73.5 (3.0) CG: *n* = 15, Age = 73.5 (4.1)	France Two Groups RCT Supervised	12 weeks, 2 × 60 min per week EG: Nintendo Wii CG: No-training no-contact	HR (monitored on 2nd, 12th, and 20th session)	EF (*n* = 5) PS (*n* = 4) VSS (*n* = 4) CP (*n* = 1)	EG > CG in *EF (*n_p_^2^* = 0.81) EG > CG in *PS (*n_p_^2^* = 0.79) EG > CG in VSS (1/4, *n_p_^2^* = 0.23) Adherence: EG (94%), CG (94%)	Unclear
Ordnung et al. (2017) [87]	Pre EG: *n* = 15, Age = 69.8 (6.3), F = n.a. CG: *n* = 15, Age = 68.6 (4.7), F = 53% Post (6 weeks) EG: *n* = 14, F = 50% CG: *n* = 15, F = 53%	Germany Two Groups RCT Supervised	6 weeks, 2 × 60 min per week EG: Xbox Kinect Video Sport Games CG: passive	Pulse (only to prevent overexertion)	EF (*n* = 2) PS (*n* = 2)	EG ↑ *PS (*r* = 0.61 and *r* = 0.55) Adherence: EG (93%), CG (100%)	low
Park and Yim (2016) [88]	Pre EG: *n* = 36, Age = 73 (3), F = 92% CG: *n* = 36, Age = 74.11 (2.9), F = 97% Post (6 weeks) EG: *n* = 36 CG: *n* = 36	Korea Two Groups RCT Supervised	6 weeks, 2 × 30 min per week EG: 1 × 30 min of conventional training, then 3D virtual Kayak Tour CG: 1 × 30 min of conventional exercise (then passive)	n.a.	MoCA	EG ↑ in MoCA CG ↓ in MoCA EG > CG in MoCA Adherence: EG (100%), CG (100%)	Low
Schättin et al. (2016) [89]	Pre EG: *n* = 15 CG: *n* = 14 Post (8 weeks) EG: *n* = 13, Age = 80, F = 38% CG: *n* = 14, Age = 80, F = 50%	Switzerland Two Groups RCT Supervised	8 weeks, 3 × 30 min per week in groups (5 min warm-up, 5 min cool-down, 20 min group specific training) EG: Video Game Dance Platform CG: Conventional Balance Training	n.a.	EF (*n* = 3) CP (*n* = 2) EEG: ERO (PFC, 1–30 Hz)	EG ↑ *EF (r = 0.46–0.57) EG ↑ *CP (r = 0.40–0.50) CG ↑ in EF (1/3, *r* = 0.39) EG ↓ Theta Power during CP Adherence: EG (87%), CG (100%)	Moderate
Schoene, et al. (2013) [90]	Pre EG: *n* = 18 CG: *n* = 19 Post (8 weeks) EG: *n* = 15, Age = 77.5 (4.5) CG: *n* = 17, Age = 78.4 (4.5)	Australia Two Groups Unsupervised RCT	8 weeks, 2–3 × 15–20 min per week EG: Video-game based Step Pad CG: Passive	n.a.	EF (*n* = 1) PS (*n* = 1) CP (*n* = 1) DT (*n* = 2)	EG > CG in CP EG > in DT (1/2) Adherence: EG (83%), CG (89%)	Low
Schoene, et al. (2015) [76]	Pre EG: *n* = 47, Age = 82 (7), F = 66% CG: *n* = 43, Age = 81 (7), F = 67% Post (16 weeks) EG: *n* = 39 CG: *n* = 42	Australia Two Groups Unsupervised RCT	16 weeks, 3 × 20 min per week EG: electronic step game CG: Passive (brochure on fall prevention)	n.a.	EF (*n* = 3) PS (*n* = 3) CP (*n* = 2) VSS (*n* =1) DT (*n* = 2)	EG > CG in PS (2/3) EG > CG in *CP EG > CG in VSS (1/2) EG > CG in DT (1/2) Adherence: EG (83%), CG (98%)	Low
Studenski et al. (2010) [93]	Pre EG: *n* = 36, Age = 80.1 (5.4), F = 83% Post (12 weeks) EG: *n* = 25, Age = 80.2 (5.4), F = 80%	USA One Group Supervised UT	12 weeks, 2 × 45–60 min per week EG: Dance Dance Revolution	n.a.	PS (*n* = 1)	Adherence: EG (70%)	moderate

Note: If reported, effect sizes are presented; * = all test measures for the respective domain were significant; d = Cohen’s effect size; r = Pearson’s correlation; n_p_^2^ = partial Eta-squared; ↑ = within group improvements or ↓decrements; > or < = group by time interaction effects; BDNF = Brain Derived Neurotrophic Factor; CG = Control Group; CP = Controlled Processing; DT = Dual-Task (motor-cognitive); EEG = Electroencephalography; EF = Executive Functions; EG = Experimental/Exergame Group; ELISA = Enzyme-Linked Immunosorbent Assay; ERO = Event Related Oscillations; ERP = Event Related Potentials; F = Female; fNIRS = functional Near-Infrared Spectroscopy; HbO2 = Oxygenated Hemoglobin; LTM = Long Term Memory; MMSE = Mini Mental State Examination; MoCA = Montreal Cognitive Assessment; n.a. = not available; PFC = Prefrontal Cortex (prefix: l = left, r = right); PS = Processing Speed; RCT = Randomized Controlled Trial; STM = Short Term Memory, UCT = Unrandomized Controlled Trial; UT = Uncontrolled Trial; VF = Verbal Fluency; VSS = Visuospatial Skill.

**Table 2 jcm-08-00734-t002:** Group by time interaction effects and within group effects for individual cognitive domains. First digit: number of significant findings, second digit: number of studies that assessed the respective domain, n.a. = no studies available, ↓ = lower performance at post-test.

	Interaction Effects		Within Group Effects		
	Active Controls	Passive Controls	Exergamers	Active Controls	Passive Controls
Executive functions	4/8	1/5	5/12	4/8 and 1/8 ↓	0/6 and 1/6 ↓
Processing speed	0/5	2/5	2/10	2/5	0/4
Controlled processing	0/1	2/3	1/4	0/1	0/3
Visuospatial skills	0/1	2/2	0/3	0/1	0/2
Verbal fluency	0/2	n.a.	0/2	0/2	n.a.
Dual-tasking	n.a.	2/2	0/2	n.a.	0/2
Short-term memory	0/3	n.a.	1/3	1/3	n.a.
Long-term memory	0/1	n.a.	1/1	1/1	n.a.
Cognitive state -MoCA	0/3	1/1	2/4	1/3	1/1↓
Cognitive state - MMSE	0/2	0/1	0/2	1/2	0/1
Overall	4/9 (44%)	4/6 (66%)	8/15 (53%)	5/9 (55%)	0/6 (0%) and 2/6↓ (33%)

Note: Some domains have been addressed by multiple measures and counted if at least one measure was significant (irrespectively of sample and effect sizes).

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
