# Peer review of "A Systematic Review on the Cognitive Benefits and Neurophysiological Correlates of Exergaming in Healthy Older Adults"

_jcm, 2019, doi:10.3390/jcm8050734_

Reviewer 1 Report

 Dear Editor,

            Thanks for the possibility to review the manuscript titled “A systematic review on the cognitive benefits and neurophysiological correlates of exergaming in healthy older adults”. I think that the manuscript addresses an issue that is very important and relevant for the researchers that study physical and cognitive training in older adults. The objectives of the review are stated clearly.

Summary

The authors start introducing the demographic change and the growing proportion of older adults with the related age-related cognitive decline. In the introduction section they also present the general neurophysiological and neuropsychological changes that affect older adults. Then they present various forms of physical and cognitive training that have been shown to attenuate age-related brain deteriorations and clearly show the aim that is to review the effects of exergame interventions  on cognitive outcomes and corresponding neurophysiological correlates in healthy older adul.

In the second phase the authors present the search strategy and show a clear flow diagram detailing the search process for selecting papers for the annotated bibliography.

On the basis of the 15 selected papers the final themes that the authors describe in detail the

potential factors contributing to exergame efficiency as well as methodological considerations of exergame systems and intervention designs

Results, limitation and clinical implication are clearly described.

 Main impressions

I think that the review addresses an important issue and it is interesting. It can be useful to describe the main problems and strengths of the training in older people, in this view the article may be useful to the knowledge base.

 1. Introduction

The introduction has a clear logical progression; it show the comparison of different non-pharmaceutical interventions. I think that to refer also to the research made with tDCS may complete this non-pharmaceutical interventions (Gangemi et al, 2018).   

 2. Method

I think that the method section is the strongest positive part of the review.

 3.  Results and discussion

I think that  the discussion of potential moderators of exergame intervention efficacy  and

neurophysiological correlates of exergaming is useful.

 Gangemi, A., Caprì, T., Fabio, R.A., Puggioni, P., Falzone, A. M. & Martino, G. (2018). Transcranial Direct Current Stimulation (tDCS) and Cognitive Empowerment for the functional recovery of diseases with chronic impairment and genetic etiopathogenesis. In Kevin V. Urbano (Ed). Advances in Genetic Research.Volume 18. New York: Nova Science Publisher.

 Originality/Novelty:       well defined

Significance:  the      results are interpreted appropriately and are significant

Quality of Presentation: the      article is written in an appropriate way? The data and analyses presented are      appropriate

Scientific Soundness: the      study is correctly designed and technically sound

Interest to the Readers: the      conclusions interesting for the readership of the Journal

Overall Merit: Yes there      is an overall benefit to publishing this work

English Level: The      English language is appropriate and understandable

 Author Response

Dear Editor, dear Reviewers,

 We did very much appreciate your reviews on our submission and would like to thank you for handling our manuscript, particularly given the short amount of time passed since the submission date. We did find your revisions very reasonable and useful and integrated respective changes according to your suggestions. In summary, we significantly revised our abstract, added a small paragraph to the methods section, improved table descriptions/notes, and adopted some parts of our discussion/conclusion. Changes are marked using the Track Changes function in Microsoft Word.

We are confident that those changes have improved the quality of our submission.

Please find our point-by-point answer below.

Point 1: The introduction has a clear logical progression; it show the comparison of different non-pharmaceutical interventions. I think that to refer also to the research made with tDCS may complete this non-pharmaceutical interventions (Gangemi et al, 2018).   

 Response 1: Thank you very much for your suggestions. We do agree that tDCS and related techniques are promising tools to improve cognitive and brain function. However, that section of our introduction was solely focused on standing-alone lifestyle approaches, which does not apply for brain stimulating methods. To not lose focus of this section, we would like to omit the suggested referencing and keep the attention on life style approaches. We, however, added the reference to our discussion, where it seemed to fit better (cf. page 20).

 Point 2: 2. Method

 I think that the method section is the strongest positive part of the review.

Point 3: 3. Results and discussion

 I think that  the discussion of potential moderators of exergame intervention efficacy  and

neurophysiological correlates of exergaming is useful.

 Response 2 and 3: Thank you for your positive feedback.

Reviewer 2 Report

Dear Authors,

I was happy to review your manuscript. This systematic review run in June 2018 investigates an important question on the potential neuropsychological benefits of exergaming on people aged 60 years or over. 

Please find my numbered comments and suggestions bellow:

Abstract

1. Optional revision – Consider structuring the abstract using a IMRaC approach.

2. Minor revision – The abstract seems very adapted for a conference but less so for a scientific article. The abstract should stand alone and make it possible to have a clear answer to the research question. The abstract would gain in clarity if the introduction was made shorter and lead to the study objective, the method include summarised points corresponding to the PRISMA checklist. The result provide quantified estimates of effects on specific outcomes, and the conclusion shortened to one or two key messages directly related to the research question.

 Introduction

3. Overall comment – The introduction is complete, up-to-date and clearly sets the theoretical framework to justify the research question. The research question is clear.

Methods

4. Overall comments – The methods is sound and adapted to answer the research question. Methods have been described in sufficient details for the study to be replicated. Classification of neuropsychological tasks as chosen outcomes is well described and argued.

5. Minor revision – A small sentence or paragraph on the analysis methods could help know what was planned. It is also not clear how effect size were measured and if only the gain over control groups were reported.

6. Optional revision – The manuscript would also gain in quality if a GRADE evaluation of the quality of evidence drawn for each outcome was added. This is however not necessary.

Results

7. Minor revision – In Table 1, in the legends, please also provide details on abbreviations used in statistical results (ex. d=Cohen's effect size comparing changes in intervention group over changes in control group). 

8. Optional revision – For readability, the table would benefit from being structured by cognitive function (one table per function) -> Not essential.

9. Minor revision – In Table 2, it might be considered that too much emphasis is put on the p-value making it difficult to assess the overall effect size. This is particularly important given most studies have very low sample size. Including indications of precision for estimates of effect size could solve this problem. Also, providing results for within groups does not provide any added values as only the between group difference is useful to assess effects of interventions. Consider removing table 2 or improving its value by providing indicators of effect size.

10. Minor revision – Only report interactions if those were planned priorly. Given the low sample size of included studies, there is bound to have apparent interactions with some secondary factors without this having much meaning (Type 1 error increased by multiple testing).

Discussion

11. Minor revision – Please make sure your discussion reflects your results. Each outcome should be assessed separately. The level of evidence is the overall effort over each outcome. If studies are not consistent and do not show similar results for similar cognitive functions, then the overall evidence of effect becomes very weak to support benefits over this function. My reading of your results is that there is no evidence of effects or absence of effects given the heterogeneity of results. However, globally, exergaming does not provide clear benefits over one function or another.

12. Overall comment – The discussion is lengthy but worth the development as it puts results into perspective with other research and provides a good overview of what is believed to take place.

Conclusion

13. Overall comment – The conclusion does not extrapolate and provides a balanced view of the results in light of what is known on the topic.

Author Response

Dear Editor, dear Reviewers,

 We did very much appreciate your reviews on our submission and would like to thank you for handling our manuscript, particularly given the short amount of time passed since the submission date. We did find your revisions very reasonable and useful and integrated respective changes according to your suggestions. In summary, we significantly revised our abstract, added a small paragraph to the methods section, improved table descriptions/notes, and adopted some parts of our discussion/conclusion. Changes are marked using the Track Changes function in Microsoft Word.

We are confident that those changes have improved the quality of our submission.

Please find our point-by-point answer below.

Point 1: Optional revision – Consider structuring the abstract using a IMRaC approach.

 Response 1: Thank you for your suggestions on our abstract’s structure, which we totally agreed with. We restructured our abstract according to the IMRaC approach.

 Point 2: Minor revision – The abstract seems very adapted for a conference but less so for a scientific article. The abstract should stand alone and make it possible to have a clear answer to the research question. The abstract would gain in clarity if the introduction was made shorter and lead to the study objective, the method include summarised points corresponding to the PRISMA checklist. The result provide quantified estimates of effects on specific outcomes, and the conclusion shortened to one or two key messages directly related to the research question.

 Response 2: Thank you for this advice. We significantly revised our abstract according to the PRISMA checklist. We draw the focus off the introduction and provide now a more detailed overview of the methods, results, and conclusion.

 Point 3: Overall comment – The introduction is complete, up-to-date and clearly sets the theoretical framework to justify the research question. The research question is clear.

 Response 3: Thank you very much. We did not make any changes to the introduction.

 Point 4: Overall comments – The methods is sound and adapted to answer the research question. Methods have been described in sufficient details for the study to be replicated. Classification of neuropsychological tasks as chosen outcomes is well described and argued.

 Response 4: Thank you, we have performed minor changes to the methods section with respect to your following suggestions.

 Point 5: Minor revision – A small sentence or paragraph on the analysis methods could help know what was planned. It is also not clear how effect size were measured and if only the gain over control groups were reported.

 Response 5: Thank you. This is a very good point that was missing in our methods section. We added a short paragraph explaining our data extraction strategy and procedure. However, due to very inconsistent information, we did not attempt to calculate effect sizes based on available statistical parameters.

Point 6: Optional revision – The manuscript would also gain in quality if a GRADE evaluation of the quality of evidence drawn for each outcome was added. This is however not necessary.

 Response 6: We did consider using a GRADE approach. However, we performed a detailed rating of the overall study quality using the Cochrane risk of bias tool that admittedly does not explicitly evaluate the certainty of effect estimates. The risk of bias tool, however, comprises a more general evaluation of the methodological study quality. We specifically wanted to show that many studies may have introduced varied sources of bias regarding insufficient intervention design and procedure. As we did not perform a meta-analysis, we intended to discuss our findings in a broader context instead of focusing too much on estimates of effect sizes.

Point 7: Minor revision – In Table 1, in the legends, please also provide details on abbreviations used in statistical results (ex. d=Cohen's effect size comparing changes in intervention group over changes in control group). 

Response 7: We have added the suggested table descriptions. Thank you for bringing this to our attention.

 Point 8: Optional revision – For readability, the table would benefit from being structured by cognitive function (one table per function) -> Not essential.

 Response 8: We considered structuring the table by cognitive functions (among other approaches) to draw the focus on the main outcomes of interest. However, we found that the readability rather suffered as many cognitive functions were addressed by a variety of studies leading to partially redundant information. Moreover, and in line with some of your further revision points, we found other study information (e.g., sample sizes, exergame systems, amount of different outcome measures per study) to be almost equally important to conceive the overall state of research on exergame interventions. If the reviewer agrees, we would like to keep our current study-wise table structure.

Point 9: Minor revision – In Table 2, it might be considered that too much emphasis is put on the p-value making it difficult to assess the overall effect size. This is particularly important given most studies have very low sample size. Including indications of precision for estimates of effect size could solve this problem. Also, providing results for within groups does not provide any added values as only the between group difference is useful to assess effects of interventions. Consider removing table 2 or improving its value by providing indicators of effect size.

 Response 9: In Table 2, we mainly aimed to present a swift and condensed overview about all findings of intervention effects on cognitive outcomes based on the strongly varying results across studies. We do agree with the reviewer that adding estimates of effect size would improve the overall value of the table. However, we were not able to comprehensively calculate effect sizes, due to inconsistencies in the reporting of statistical parameters in several studies, which may mislead interpretations if reported inconsequently. Nevertheless, we slightly adopted the table description and added relative values to overall summed values for a better comprehension. Information on within group effects were added to give a further impression on, for example, the influence of potential retest effects and the general sensitivity to training effects of individual cognitive domains. However, if the reviewers insists, we would agree to remove table 2 or the within group effects section of the table.

Point 10: Minor revision – Only report interactions if those were planned priorly. Given the low sample size of included studies, there is bound to have apparent interactions with some secondary factors without this having much meaning (Type 1 error increased by multiple testing).

Response 10: Thank you for that relevant suggestion. However, we did not feel qualified enough to judge whether interactions where planned a priori. Therefore, we focused to report all stated interaction effects. We did, nevertheless, discuss that insufficient statistical approaches and multiple testing of executive functions may have contributed to versatile findings and type I error (see chapter 4.2 paragraph one and two).

 Point 11: Minor revision – Please make sure your discussion reflects your results. Each outcome should be assessed separately. The level of evidence is the overall effort over each outcome. If studies are not consistent and do not show similar results for similar cognitive functions, then the overall evidence of effect becomes very weak to support benefits over this function. My reading of your results is that there is no evidence of effects or absence of effects given the heterogeneity of results. However, globally, exergaming does not provide clear benefits over one function or another.

Response 11: Thank you for bringing this to our attention, we appreciate your suggestion. We do agree with your appraisal and revised our discussion accordingly, leading to a more consistent overall interpretation and summary of our findings.

We agree with the reviewer that exergaming does not provide clear evidence for improvements on distinct cognitive functions and hence should be described respectively. Please do also note our critical summaries, for example in chapter 4.2 second paragraph: “[…] might indicate that exergames have no further advantages over traditional exercise”.

Point 12: Overall comment – The discussion is lengthy but worth the development as it puts results into perspective with other research and provides a good overview of what is believed to take place.

Response 12: Thank you very much. We further improved our discussion according to your recommendations.

Point 13: Overall comment – The conclusion does not extrapolate and provides a balanced view of the results in light of what is known on the topic.

Response 13: Thank you. After slightly adjusting our discussion, we somewhat adopted our conclusion according to your suggestions as well.
